

# Impact of exposure to tobacco smoke, arsenic, and phthalates on locally advanced cervical cancer treatment—preliminary results

Iulia A. Neamtiu[1,2], Michael S. Bloom[1,3], Irina Dumitrascu[4,5], Carmen A. Roba[6], Cristian Pop[4,5,6], Claudia Ordeanu[7], Ovidiu Balacescu[8] and Eugen S. Gurzau[1,2,9]

[1] Health Department, Environmental Health Center, Cluj-Napoca, Romania
[2] IMOGEN Research Institut, Cluj-Napoca, Romania
[3] Departments of Environmental Health Sciences and Epidemiology and Biostatistics, University at Albany, State University of New York, Rensselaer, NY, United States of America
[4] Physico-chemical and Biotoxicological Analysis Laboratory, Environmental Health Center, Cluj-Napoca, Romania
[5] Cluj School of Public Health - College of Political, Administrative and Communication Sciences, Babes-Bolyai University, Cluj-Napoca, Romania
[6] Faculty of Environmental Science and Engineering, Babes-Bolyai University, Cluj-Napoca, Romania
[7] Radiotherapy II Department, The Oncology Institute "Prof. Dr. Ion Chiricuta", Cluj-Napoca, Romania
[8] Functional Genomics, Proteomics and Experimental Pathology Laboratory, The Oncology Institute "Prof. Dr. Ion Chiricuta", Cluj-Napoca, Romania
[9] University of Medicine and Pharmacy "Iuliu Hatieganu", Cluj-Napoca, Romania

Corresponding author
Iulia A. Neamtiu, iulianeamtu@ehc.ro

## ABSTRACT

**Background**. Cancer research is a national and international priority, with the efficiency and effectiveness of current anti-tumor therapies being one of the major challenges with which physicians are faced.

**Objective**. To assess the impact of exposure to tobacco smoke, arsenic, and phthalates on cervical cancer treatment.

**Methods**. We investigated 37 patients with locally advanced cervical carcinoma who underwent chemotherapy and radiotherapy. We determined cotinine and five phthalate metabolites in urine samples collected prior to cancer treatment, by gas chromatography coupled to mass spectrometry, and urinary total arsenic by atomic absorption spectrometry with hydride generation. We used linear regression to evaluate the effects of cotinine, arsenic, and phthalates on the change in tumor size after treatment, adjusted for confounding variables.

**Results**. We detected no significant associations between urinary cotinine, arsenic, or phthalate monoesters on change in tumor size after treatment, adjusted for urine creatinine, age, baseline tumor size, and cotinine (for arsenic and phthalates). However, higher %mono-ethylhexyl phthalate (%MEHP), a putative indicator of phthalate diester metabolism, was associated with a larger change in tumor size ($\beta = 0.015$, 95% CI [0.003–0.03], $P = 0.019$).

**Conclusion**. We found no statistically significant association between the urinary levels of arsenic, cotinine, and phthalates metabolites and the response to cervical cancer treatment as measured by the change in tumor size. Still, our results suggested that

phthalates metabolism may be associated with response to treatment for locally advanced cervical cancer. However, these observations are preliminary and will require confirmation in a larger, more definitive investigation.

## INTRODUCTION

Cervical cancer is among the leading causes of cancer-related morbidity in women worldwide; in 2012 there were 4,343 new cases in Romania alone, accounting for 1,909 deaths (*Ferlay et al., 2013*). Response to treatment consisting of radiotherapy combined with cisplatin-based chemotherapy in late stages, in most cases of locally advanced cervical cancers, is good, however, 5-year free survival rates are unsatisfactory (*Eifel et al., 2009*).

Human papilloma virus (HPV) infection (particularly types 16 and 18) is present in more than 90% of cervical cancer cases (*Bosch et al., 1995*); however, not all HPV infections result in cervical cancer (*Burd, 2003*). In addition to HPV infection, evidence from epidemiologic studies also supports an association between active cigarette smoking and cervical neoplasia (*Haverkos et al., 2003*; *Trimble et al., 2005*). Even in non-smokers, lifetime tobacco exposure may contribute to the development of pre-cancerous cervical intraepithelial neoplasia (*Wu et al., 2003*).

Exposure to inorganic arsenic has also raised concern worldwide, as it has been associated with lung, bladder, and skin cancer development, among others (*Smith et al., 1992*). Arsenic induces higher levels of reactive oxygen species (*Flora, 2011*), an excess of which peroxidizes lipids and oxidizes proteins and nucleic acids, further causing DNA lesions (*Ercal, Gurer-Orhan & Aykin-Burns, 2001*). A growing body of evidence indicates that exposure to arsenic may impair the immune system by suppressing *T* and *B* lymphocyte maturation, inducing apoptosis of macrophages and lymphocytes, and impeding *T* cell specific cytokine expression (*Vega et al., 1999*; *Cheng et al., 2004*), thus promoting cancer development. Arsenic triggered oncogenesis also leads to immune suppression, further facilitating tumor growth and generating a positive feedback loop between cancer and immune function (*Acharya et al., 2010*) that may also impact the effectiveness of treatment.

Phthalates are widely used in many consumer goods, facilitating frequent human exposures (*Kamrin, 2009*). Phthalates are endocrine disruptors that have been associated with adverse health effects especially as a result of early life exposures (*Braun, Sathyanarayana & Hauser, 2013*), exhibiting modulation actions on gonadal steroids that might result in female reproductive health outcomes (*Kay, Chambers & Foster, 2013*). Importantly, phthalates also appear to play a role in inflammation, which at a chronic level precedes tumorigenesis (*Anand et al., 2008*). Phthalates-mediated increases in chronic inflammation have been demonstrated in the prostate, uterus, ovary, and breast, all common locations for neoplastic proliferation (*Singh & Li, 2012*).

While environmental factors are likely to contribute as component causes in the etiology of cancer (*Wu et al., 2016*), there appears to be little if any data available to assess effects of environmental pollutants on cancer treatment. Still, it is plausible to speculate that environmental contaminants such as tobacco smoke, arsenic, and phthalates may interfere with cancer treatment. Contaminant-mediated inflammation might modify the immune response to neoplastic proliferation (*Grivennikov, Greten & Karin, 2010*), affecting tumor development and progression, and amplifying (*Zitvogel et al., 2008*) or decreasing (*Ammirante et al., 2010*) the response to therapy. Furthermore, these contaminants may impact individual detoxifying capacity and thus, the effectiveness of pharmaceutical agents used for treatment. For example, hepatic cytochrome P450 monooxygenases (CYP450) govern transformation of phthalate diesters to their metabolites (*Greenblatt et al., 2002*; *Frederiksen, Skakkebaek & Andersson, 2007*), the activity of which was decreased in association with exposure to di(2-ethylhexyl) phthalate (DEHP) in a rat model (*Parmar, Srivastava & Seth, 1994*). This fact might alter CYP450-dependent metabolic processes for Taxol (one of the chemotherapeutics) (*Rowinsky et al., 1993*) among DEHP-exposed women. In addition, the investigated contaminants may interact with the chemotherapeutics by competing for binding to carriers/receptors or by producing alteration in tumor tissue or side effects requiring reduction of the therapeutic dose. Both arsenic and total platinum (one of the blood species of the therapeutic agent, carboplatin) bind to proteins (even that protein binding is limited in case of platinum (*Van der Vijgh, 1991*)), and as arsenic binds to protein thiol or sulfhydryl groups, it may act as a competitor for platinum. In fact, clinical trials showed therapeutic value for arsenic trioxide in acute promyelocytic leukemia (*Soignet et al., 2001*) and myeloma (*Deaglio et al., 2001*) and few recent studies pointed out that it can produce apoptosis via alterations in specific cell signaling pathways (*Seol et al., 1999*; *Alemany & Levin, 2000*). Finally, response to cervical cancer therapy may differ by its etiology and moreover, the therapy response may differ due in part to environmental contaminants contributing as component causes to cases, including non-HPV confirmed cases, from those attributable solely to HPV infection. As part of a larger investigation into the factors that modify response to treatment among women with invasive cervical cancer, we conducted an interim, exploratory analysis to assess the impact of exposures to tobacco smoke, arsenic, and phthalates. To the best of our knowledge, no prior investigations have assessed the impact of these widely distributed environmental pollutants on the effectiveness of treatment for locally advanced cervical cancer.

# MATERIALS AND METHODS

## Study population

Our study population comprised women receiving treatment for locally advanced cervical cancer at the Oncology Institute "Prof Dr Ion Chiricuta" (Cluj-Napoca, Romania), between 2013 and 2014. Patients were eligible for participation if: (1) diagnosed with histologically-confirmed squamous cell carcinoma of the cervix at clinical stages IIB, IIIA, or IIIB according to the MD Anderson Cancer Center modification of the International

Federation of Gynecology and Obstetrics (FIGO) diagnostic criteria (*Benedet et al., 2000*); (2) aged 20–79 years at the time of diagnosis; (3) diagnosed with Zubrod score $\leq 2$ (an overall well-being index scored as 0–5, for which $0 =$ asymptomatic and $5 =$ death) (*Oken et al., 1982*); (4) blood hemoglobin $\geq 9$ g/dL, leucocytes $\geq 3,000/mm^3$, and platelets $\geq 100,000/mm^3$; (5) urine creatinine $< 1.2$ mg/L and urine nitrogen $< 80$ mg/L; and (6) normal transaminases. We excluded patients with: (1) a history of a prior malignancy, including previous cervical cancer; (2) interrupted treatment (women who stopped therapy for any reason); or (3) cardiovascular, kidney, or liver function that was deemed too poor to initiate treatment. Of the eligible patients contacted (44 patients), 97.7% (43 patients) agreed to be enrolled in our study. All participants provided written informed consent prior to study participation and the research protocol was approved by the Ethics Committee in Research and Development and Quality Assurance for Clinical Studies at the Oncology Institute "Prof. Dr. Ion Chiricuta" (approval stated in the Ethics Committee Evaluation Report no. 6490/2013).

All study participants underwent three cycles of chemotherapy with Taxol and Carboplatin (AUC5) followed by radiotherapy (all study participants received the same doses of radiation and chemotherapeutic drugs), according to the usual clinical cervical cancer treatment protocol, as previously described in detail (*Balacescu et al., 2014*). We assessed treatment response as the difference in tumor size measured before and after radio and chemotherapy, using a computerized tomography scan. Treatment outcome data were available for 37 (86%) enrolled participants. Six study participants were still undergoing treatment at the time when the statistical analysis on this preliminary data was done and so outcome data (tumor size after treatment) were not available.

## Urine samples collection and analysis

We analyzed one urine sample, collected prior to cancer treatment, for each study participant. Study nurses collected urine specimens at the time of the cancer diagnosis, into 50 mL polyethylene containers previously decontaminated with nitric acid and then rinsed with water. Within 15 min of urine collection, samples were frozen at –20 °C and then transferred to the Environmental Health Center (Cluj-Napoca, Romania), where they were stored until analysis for cotinine, arsenic, and five phthalate monoesters. Urinary creatinine was measured according to a previously described procedure, (*Neamtiu, Dumitrascu & Roba, 2014*), for which the intra-assay coefficient of variation (CV) was 6.5%.

### *Urinary cotinine*

The analytic method for determination of urinary cotinine was previously described in detail (*Neamtiu, Dumitrascu & Roba, 2014*). In brief, cotinine was extracted with dichloromethane, dissolved in toluene, and analyzed using a QP 2010 Plus NCI gas chromatograph (Shimadzu, Japan) coupled to a mass spectrometer (GC-MS) operated in the selective ion monitoring mode. The method limit of detection (LOD) was 10 µg/L and the intra-assay CV was 3.47%.

### Urinary arsenic

The total arsenic concentration in urine samples was analyzed using a Zeenit 700P atomic absorption spectrometer with hydride generation system (Analytik Jena, Germany). To determine arsenic, 5 mL of urine was mixed with 5 mL $HNO_3$ and 2 mL $H_2O_2$ and mineralized using a Mars 6 microwave digestor (CEM Corporation, Matthews, NC, USA). The mineralized sample was then diluted to 25 mL with ultrapure water. The mineralized sample reacts with sodium borohydride in an acid environment and forms volatile metal hydrides, which were atomized in a quartz cell heated at 960 °C. After plotting the calibration curve (arsenic specific wavelength, $\lambda = 193.7$ nm), the processed samples were atomized and their absorbencies were measured. The method LOD was 0.5 μg/L and the intra-assay CV was 2.36%.

### Urinary phthalates

We determined urinary phthalate monoester metabolites of dibutyl phthalate (DBP), including mono-butyl phthalate (MBP), benzyl-butyl phthalate (BzBP), including mono-benzyl phthalate (MBzP), and di-ethylhexyl phthalate (DEHP), including mono-(2-ethylhexyl) phthalate (MEHP), mono-(2-ethyl-5-hydroxyhexyl) phthalate (MEHHP), and mono-(2-ethyl-5-oxo-hexyl) phthalate (MEOHP), based on a recently published method (*Kim et al., 2014*). Briefly, after enzymatic (*β*-glucuronidase) hydrolysis of urine samples, phthalate metabolites were extracted with a solvent mixture (hexane, acetone) by sonication. The organic phase was dried and evaporated to dryness. A derivatization agent, N,O-bis(trimethylsilyl) trifluoroacetamide with trimethylchlorosilane 1% (BSTFA with TMCS 1%), was added, and the samples were kept in a thermoreactor (Techne, Cambridge, UK) at 65 °C for 1 hour. For the analysis, we used a GC-MS QP 2010 Plus NCI (Shimadzu Corporation) in the single ion monitoring mode. The method LOD was 2.5 μg/L and intra-assay CVs were 9.64% (MBP), 9.01% (MBzP), 5.43% (MEHP), 8.7% (MEOHP), and 4.78% (MEHHP).

## Data analysis

We characterized distributions for exposures and covariates, and imputed urinary cotinine, arsenic, and phthalates values below the method LODs as LOD/$\sqrt{2}$ prior to analysis (*Hornung & Reed, 1990*). We also calculated %MEHP as [MEHP/(MEHP + MEOHP + MEHHP) × 100] on a molar basis, to assess the impact of phthalates metabolism (*Hauser, 2008*). To evaluate the unadjusted effects of environmental exposures on response to cervical cancer therapy, defined as the difference in tumor size before and after treatment, we used a series of individual regression models, including only urine creatinine as a covariate in addition to either cotinine, arsenic, or each phthalate as the sole predictor. We also constructed a series of comprehensive multiple linear regression models to evaluate the impact of confounder-adjusted environmental exposures on cervical cancer treatment response, including age, baseline tumor size, and urine cotinine (arsenic and phthalates models) as covariates.

Given our use of spot collections for biomarkers of exposure, we adjusted for diurnal variation in urine volume by including creatinine as a covariate in the regression models

(*Kim et al., 2011*). However, in a second set of regression models, we also used a more 'traditional' creatinine correction, in which urinary cotinine, arsenic, and phthalates were divided by urine creatinine and the 'normalized' variables entered into the regression models. We examined the distribution of residuals from all regression models to verify the tenability of the normality assumption and to identify outlying and influential observations for further examination. Stata v.12 (StataCorp LP, College Station, TX USA) was used for the statistical analysis, and statistical significance was defined as $p < 0.05$ for a two-tailed test.

## RESULTS

As described in Table 1, 37 participants were 52 years of age on average at the time of the cervical cancer diagnosis (range 26–76 years). Approximately 59% ($n = 22$) of patients responded well to the cancer treatment, presenting an 80–100% reduction of the initial tumor size, although the tumor size was reduced by less than 80%, in 41% of women. Most participants (59%) were non-smokers (including 13 women who reported tobacco exposure), six were former smokers (16%), and 9 (24%) women self-identified as smokers.

Table 2 shows the distributions for measured urinary cotinine, arsenic, and phthalates levels on a creatinine basis. Values are reported on a wet-weight basis in Table S1. Urinary cotinine values ranged from <LOD-395.4 μg/g creatinine with a geometric mean of 13.9 μg/g creatinine. Total arsenic values ranged from <LOD-115.4 μg/g creatinine with a geometric mean of 13.1 μg/g creatinine. Using ANOVA, cotinine was higher in smokers compared to non-smokers ($p = 0.04$) and to former smokers ($p = 0.08$). On a creatinine basis, geometric mean values for 35 women with sufficient urine volume available for phthalates determination were 8.8 μg/g MBP, 5 μg/g MBzP, 15.8 μg/g MEHP, 3.5 μg/g MEOHP, and 8.7 μg/g MEHHP. The highest maximum values were measured for MBP (295.5 μg/g) and MBzP (182.9 μg/g), whereas the other phthalates metabolites had lower maximum levels: MEHP (91.9 μg/g), MEOHP (34.1 μg/g), and MEHHP (88.6 μg/g). The geometric mean %MEHP was 68.2, ranging from a minimum value of 10.7% to a maximum value of 97.2%.

Table 3 describes the multiple linear regression analysis of cervical cancer treatment response, measured as the change in tumor size, upon environmental exposures, adjusted for confounding variables. All effect estimates were of small magnitude and 95% confidence intervals (95% CI) included the null hypothesis; we detected no statistically significant confounder-adjusted associations between urinary cotinine, total arsenic, or phthalate metabolites and the change in tumor size. We repeated the models without adjusting for cotinine, with no meaningful change in results (data not shown). In contrast, the results for the unadjusted regression analysis of total arsenic were statistically significant ($\beta = 0.01$, 95% CI 0.0003, 0.02; $P = 0.045$), as was an age-only adjusted regression analysis ($\beta = 0.012$, 95% CI 0.002, 0.023; $P = 0.025$), although results were similarly null in the fully adjusted model. The results were also similar when we used a traditional creatinine correction procedure in lieu of adjustment for urinary creatinine as

**Table 1  Demographic and clinical characteristics of women receiving cervical cancer treatment and participating in the study ($n = 37$).**

| Characteristics | Mean (n) | SD (%) | Min. | 50th %tile | Max. |
|---|---|---|---|---|---|
| Age (years) | 52.2 | 11.1 | 26 | 54 | 76 |
| Body mass index (kg/m$^2$) | 27.1 | 5.4 | 17 | 26.8 | 41.2 |
| Smoker (yes) | (9) | (24.3) | – | – | – |
| Tumor Stage | | | | | |
| IIB | (13) | (35.2) | – | – | – |
| IIIA | (14) | (37.8) | – | – | – |
| IIIB | (10) | (27) | – | – | – |
| Change in tumor size (cm) | 3.3 | 1.3 | 1 | 3 | 6 |

**Table 2  Urinary cotinine, arsenic, and phthalates metabolites measured in $n = 37$ women undergoing cervical cancer treatment and participating in the study ($\mu$g/g creatinine).**

| Analyte | Min. | 25th %tile | 50th %tile | 75th %tile | Max. | Geometric mean |
|---|---|---|---|---|---|---|
| Cotinine | <LOD | <LOD | 9.3 | 35.3 | 395.4 | 13.9 |
| Arsenic | <LOD | 7.4 | 13.2 | 20.8 | 115.4 | 13.1 |
| MBP | <LOD | <LOD | 9.8 | 22 | 295.5 | 8.8 |
| MBzP | <LOD | 1.8 | 5.4 | 10.7 | 182.9 | 5 |
| MEHP | 1.5 | 10.6 | 15.5 | 28.3 | 91.9 | 15. 8 |
| MEOHP | <LOD | <LOD | 2.8 | 8.2 | 34.1 | 3.5 |
| MEHHP | <LOD | 4.6 | 8.8 | 13.8 | 88.6 | 8.7 |
| %MEHP | 10.7 | 63.6 | 85.4 | 91.6 | 97.2 | 68.2 |

**Notes.**
LOD, method limit of detection; MBP, mono butyl phthalate; MBzP, mono benzyl phthalate; MEHP, mono (2-ethylhexyl) phthalate; MEOHP, mono (2-ethyl-5-oxohexyl) phthalate; MEHHP, mono (2-ethyl-5-hydroxyhexyl) phthalate; %MEHP, $100 \times$ (MEHP/(MEHP + MEOHP + MEHHP) on a molar basis.

a covariate in regression models (Table S2). However, as described by Fig. 1, we detected a statistically significant positive association between %MEHP and response to cervical cancer therapy, in the multivariable regression model adjusted for confounders ($\beta = 0.015$; 95% CI [0.003–0.03]; $P = 0.019$). However, %MEHP results were not statistically significant in an unadjusted model ($\beta = 0.004$; 95% CI [$-0.01$–0.02]; $P = 0.634$) or adjusted for age only ($\beta = 0.012$; 95% CI [$-0.007$–0.031]; $P = 0.205$).

## DISCUSSION

We evaluated the impact of exposure to tobacco smoke, arsenic, and phthalates on locally advanced cervical cancer treatment using urine biomarkers from 37 women aged 26–76 years. More than a half of the study participants responded well to cancer treatment, presenting a reduction of the initial tumor size between 80% and 100%. We found no meaningful effects on the change in tumor size for urine cotinine or total arsenic, or for five monoester metabolites of three widely distributed phthalate diesters. Yet, our analysis of %MEHP suggested an enhanced response to cervical cancer treatment, indicating that less efficient conversion of the primary hydrolytic DEHP phthalate monoester MEHP, to

**Table 3** Associations for urinary cotinine, arsenic, and phthalates (μg/l) with cervical cancer therapy response, adjusted for covariates using multiple linear regression models.[a]

| Predictors | n | β | 95% | CI | P-value |
|---|---|---|---|---|---|
| Cotinine | 37 | −0.001 | −0.005 | 0.003 | 0.590 |
| Arsenic | 37 | 0.005 | −0.002 | 0.013 | 0.173 |
| MBP | 35 | 0.0003 | −0.001 | 0.002 | 0.752 |
| MBzP | 35 | −0.001 | −0.013 | 0.010 | 0.831 |
| MEHP | 35 | 0.0007 | −0.006 | 0.007 | 0.817 |
| MEOHP | 35 | −0.016 | −0.044 | 0.012 | 0.256 |
| MEHHP | 35 | 0.0006 | −0.006 | 0.007 | 0.855 |
| **%MEHP[b]** | **35** | **0.015** | **0.003** | **0.03** | **0.019** |

**Notes.**

[a] Adjusted for baseline tumor size (cm), age (years), urinary creatinine (mg/l), and urinary cotinine for arsenic and phthalates (μg/l).

[b] %MEHP = $100 \times$ (MEHP/(MEHP + MEOHP + MEHHP)) on a molar basis.

MBP, mono butyl phthalate; MBzP, mono benzyl phthalate; MEHP, mono (2-ethylhexyl) phthalate; MEOHP, mono (2-ethyl-5-oxohexyl) phthalate; MEHHP, mono (2-ethyl-5-hydroxyhexyl) phthalate.

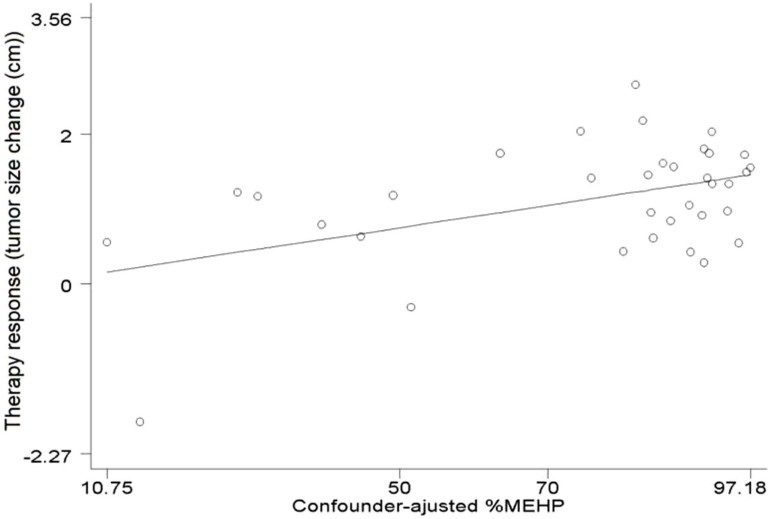

**Figure 1 Association between %MEHP and response to cervical cancer treatment.** Adjusted for initial size of the tumor, urine cotinine, creatinine, and age %MEHP = $100 \times$ (MEHP/(MEHP + MEOHP + MEHHP)) on a molar basis.

its secondary-oxidative metabolites MEOHP and MEHHP, may confer benefit (*Hauser, 2008*). It is tempting to speculate the existence of a common metabolic pathway impacting hydrolysis and oxidation of DEHP and response to the cervical cancer treatment protocol. Contaminant-associated modulation of normal metabolic processes (e.g., phthalates) is also likely to impact the pharmacokinetics of chemotherapeutic drugs (*Greenblatt et al., 2002*). Reduced drug metabolism might result in a longer half-life of the drug, increasing the potential for dose related adverse health effects. In contrast, accelerated drug metabolism might result in a shorter half-life of the drug, reducing its effectiveness. However, given the limited sample size available for our study, and the absence of

more comprehensive indicators to characterize cervical cancer therapy response, these observations should be considered preliminary and will require confirmation in a more definitive investigation.

Several human biomonitoring studies from nearby European areas reported urinary levels of cotinine, arsenic, and phthalates measured in general population samples. Geometric mean urinary cotinine was modestly lower in 120 Romanian mothers 25–45 years of age (9.10 µg/g creatinine) (*Lupsa et al., 2015*) than for women in our study. However, the geometric mean total urine arsenic value for 4,730 Germans aged 18–69 (3.1 µg/g) was substantially lower than for women in our study (*Becker et al., 2003*). In our study, we determined urinary metabolites of DBP, BzBP, and DEHP as those phthalates most prevalent in Central and Eastern European populations (*Černá et al., 2015*). The highest urinary phthalates values in our study population were measured for MBP, MBzP, MEHP and MEHHP, while MEHP had the highest geometric mean. The geometric mean concentrations of MBzP and MEHP were higher in our study than were reported for 117 women from the Czech Republic (4.35 and 3.13 µg/g creatinine, respectively), 125 women from Slovakia (3.81 and 3.22 µg/g creatinine, respectively), and 115 women from Hungary (3.82 and 3.70 µg/g creatinine, respectively) (*Černá et al., 2015*). In contrast, levels of MEOHP and MEHHP were lower in our study, than reported for Czech (11.68 and 18.45 µg/g creatinine, respectively), Slovakian (11.54 and 18.20 µg/g creatinine, respectively), and Hungarian (10.37 and 15.54 µg/g creatinine, respectively) women (*Černá et al., 2015*). Overall, the distribution of exposures among our study population was unique, yet we identified similarities with values reported for several nearby, non-clinical European populations.

Our study had several important limitations, and so, our results should be interpreted only as hypothesis generating. The small number of study participants may have limited our ability to detect modest associations, and it precluded an analysis of multiple exposures included in a single regression model. We hope to expand the sample size in the future to generate more precise effect estimates and to more comprehensively analyze exposure to the mixture of tobacco smoke, arsenic, and phthalates. While our study was prospective in nature, we measured exposure at only a single time point prior to initiation of cancer treatment; the short *in vivo* 1/2-lives and the episodic nature of phthalates exposure may have misclassified some patients (*Fromme et al., 2007*). Our use of a total arsenic variable, including comparatively innocuous organic species with toxic inorganic species, may have further misclassified exposure for some women (*Marchiset-Ferlay, Savanovitch & Sauvant-Rochat, 2012*). Yet, misclassification is unlikely to have differed by study outcome and so any bias will have led to underestimated effects. Finally, we did not incorporate recent data suggesting an important role for gene expression on cancer treatment response in our study population (*Balacescu et al., 2014*). Still, we do not anticipate a link to exposure and so bias was unlikely.

To the best of our knowledge, this preliminary report describes the first investigation on the impact of widely distributed environmental contaminants on the effectiveness of locally advanced cervical cancer therapy. These results should help to reassure clinicians that even levels of cotinine, total arsenic, MBzP, and MEHP higher than reported from

other European study populations are unlikely to interfere with the effectiveness of radio and chemotherapy for invasive cervical cancer. Still, %MEHP may prove important. Susceptibility to therapy might still differ substantially by exposure status and by HPV status in patients diagnosed with advanced cervical cancer, so we plan to stratify the analysis by HPV status in an expanded future study sample. In this particular case, a linear regression overall probably would not show any strong effects, but among a small group of patients the exposure status might still be a valuable predictor of therapy effectiveness.

In a larger future investigation, we will incorporate longitudinal collection of urine specimens during cancer treatment to reduce exposure misclassification as well as gene expression information, to assess the impact on treatment response. We also plan to evaluate the impact of environmental contaminants (cotinine, arsenic, and phthalates) on post transcriptional targets (micro RNA (miRNA)), in the context of HPV-type, to delineate prognostic indicators for treatment effectiveness of advanced stage (IIB–IIIB) cervical cancer.

Current cancer treatments are both expensive and induce serious side effects, and so characterizing the potential impact of widely distributed environmental pollutants is critical to establish new indicators for predicting the effectiveness of cervical cancer treatments.

## CONCLUSIONS

Exposure to tobacco smoke, arsenic, and phthalates did not appear to impact cervical cancer treatment at the levels of exposure experienced by our study population. Phthalates metabolism may be associated with locally advanced cervical cancer treatment response, although the clinical relevance is unclear. A more comprehensive investigation with a larger sample size will be necessary for a more definitive result.

## ACKNOWLEDGEMENTS

The authors would like to thank the study participants, without whom this investigation would not have been possible.

### Funding
The financial support for this research was provided by the UEFISCDI Program - PN-II-PT-PCCA-2011-32-1328 (grant no 96/2012). The funders had no role in study design, data collection and analysis, decision to publish, or preparation of the manuscript.

### Grant Disclosures
The following grant information was disclosed by the authors:
UEFISCDI Program - PN-II-PT-PCCA-2011-32-1328: 96/2012.

### Competing Interests
The authors declare there are no competing interests.

## Author Contributions

- Iulia A. Neamtiu analyzed the data, wrote the paper, prepared figures and/or tables.
- Michael S. Bloom analyzed the data, wrote the paper, prepared figures and/or tables, reviewed drafts of the paper.
- Irina Dumitrascu and Cristian Pop performed the experiments, contributed reagents/-materials/analysis tools, reviewed drafts of the paper.
- Carmen A. Roba and Claudia Ordeanu contributed reagents/materials/analysis tools, reviewed drafts of the paper.
- Ovidiu Balacescu and Eugen S. Gurzau conceived and designed the experiments, reviewed drafts of the paper.

## Human Ethics

The following information was supplied relating to ethical approvals (i.e., approving body and any reference numbers):

Ethics Committee in Research and Development and Quality Assurance for clinical studies of the Oncology Institute ''Prof. Dr. Ion Chiricuta''; Ethics Committee Evaluation Report no. 6490/2013.

## Data Availability

The raw data has been supplied as a Supplementary File.

## Supplemental Information

Supplemental information for this article can be found online at http://dx.doi.org/10.7717/peerj.2448#supplemental-information.

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
