# Peer review of "Impact of exposure to tobacco smoke, arsenic, and phthalates on locally advanced cervical cancer treatment—preliminary results"

_PeerJ, doi:10.7717/peerj.2448_

## Round 0.1 · original submission · Major Revisions

Please address all issues raised by reviewers. please include a point by point response to each issue listing how the issue was addressed and where it was addressed in the revised manuscript.

·

Basic reporting

This is generally a well written paper dealing with an important and valid medical issue. I do have some rather minor comments regarding the language:
Line 52: “approximately 4,343 new cases”: this looks a very precise number. So the word “approximately” sounds queer. Maybe the authors want to say that it is “at least 4,343” cases (indicating, that there might be more cases than reported by Ferlay)?
Line 206ff: This sentence is simply incomplete! (“whereas maximum values for xy ().”)

Experimental design

Apart from these minor comments regarding language I do have at least one major concern: The authors do explain in their introduction that the chosen environmental substances do pose a cancer risk and they also provide some rationale why this might be the case and provide some evidence. Then they report that (line 90f) “there appears to be little if any data available to assess effects of environmental pollutants on cancer treatment.” This might well be true. But still I would rather the authors put forward one or more hypotheses how environmental toxicants might affect treatment. In my mind there are several possibilities: (1) the cancer causing activities of the substances (mutagenic properties, inflammation, immune modulation,…. – by the way: the authors forgot to mention the endocrine disruptive effects of phthalates! – are still acting on the cancer tissue partly setting off the treatment effects. In that case the outcome would simply be an effect of the difference of the tumor reducing effect of the therapy and the tumor progressing effect of the toxicants. (2) The biomonitoring data do not only indicate external exposure but also the detoxifying capacity of the body due to the person’s genetic composition. This capacity could also have an impact on the pharmaceutical agents. This hypothesis is shortly discussed by the authors later. (3) The toxics directly interact with the therapy (e.g. competing at receptors or causing a change in the metabolic state in the tumor tissue or increasing the risk of side effects so that the dose of the therapeutics must be reduced etc.). (4) There are several causes of cervical cancer. As the authors rightfully explain HPV is the main cause but not the only one. Maybe with higher exposure to a cancer causing agent it is more likely that this agent was the cause of the very cancer. Maybe cervical cancer differs (in susceptibility to therapy) depending on its cause and origin. Maybe a cancer caused by virus infection is less or more strongly affected by the standard therapy than a cancer caused by smoking?
I am not sure if the last option is very likely. But it would open up the possibility that there is not effect of exposure seen in the 90% of cancers caused by HPV but among the 10% not caused by HPV susceptibility to therapy might still differ substantially by exposure status. In that case a linear regression overall would not show any strong effects. But among a small group of patients the exposure status might still be a valuable predictor of therapy effectiveness. I do understand that sub-group analysis is no option with so few cases only. But at least in the discussion the authors should point the ways forward (e.g. analyzing by HPV status or by hormonal receptor status).
So this is my main suggestion: In the introduction to explain the main hypotheses behind the study and in the discussion to suggest in more detail how to proceed from there on.

Validity of the findings

no comment

Additional comments

Some minor points content wise are:
Line 61: This study uses cotinine as an exposure marker. This marker is increased not only from ETS but even more so by active smoking. I do agree that also ETS (alone) could be a risk factor. But why stress this point so much when the own study includes active smokers (who in fact do display the highest cotinine levels and thus would drive any association found in the regression analysis)?
Line 100 and later: I am not sure what “recruited” means. (Language is ambiguous!) Did they approach 43 women of which 97.7% (i.e. 42) consented to participate or did they approach 44 of which 43 (i.e. 97.7 of the total) were finally recruited?
Line 110ff: interrupted treatment served as exclusion criterion. But did all women receive the same dose? (See e.g. my 3rd hypothesis of toxic exposures increasing the risk of side effects!) In general we learn nothing about dose of therapeutic agents!
Line 120ff: I do not fully understand the issue of surgical treatment. Any surgical treatment would substantially reduce the tumor mass. How was this accounted for? Or were the women receiving surgical treatment excluded from the analysis? This would account for finally only including 37 women in the analysis. (Line 124) I cannot think of any other reason for missing these vital outcome data!
Line 177: As confounders age, cotinine and baseline tumor size were considered. A parameter acts as a confounder if it is related with the exposure and the outcome. For example including cotinine in the adjusted model would not make any sense when it is not associated with tumor reduction in the simple regression analysis. Adding an unnecessary confounder when the number of data points is so small is a very dangerous and usually unnecessary act! Even if a parameter is associated with both exposure and outcome it is not necessarily a confounder. It could also act as effect intermediate factor mediating the association between exposure and outcome. This mostly concerns tumor size. This consideration is important as the authors found an effect of arsenic in the unadjusted but not in the adjusted model. It would be interesting to see which assumed confounder caused the association to disappear. On the other hand it is surprising that %MEHP only showed an effect in adjusted but none in unadjusted analysis. Again which confounder caused this change in outcome? I do find it rather difficult to discuss an effect that is not also to some extend evident in the raw data (unadjusted model). How would such an effect be clinically meaningful?

Reviewer 2 ·

Basic reporting

in reviewed manuscript Authors analysed environmental exposure to arsenic, phthalates and tobacco smoke in relation locally advanced cervical cancer treatment.
In my opinion the idea that exposure in above factors in the past have impact on the progression of treatment is incorrect. These factors have impact on cancer development but in was no simple relations: eg. phthalates are endocrine disrupting chemicals and the way from phthalates exposure via changing hormonal metabolism to cancer development is very long; another mechanism are involved in arsenic induced carcinogenesis - it is observed increased oxidative stress active biomarkers, leading to chromosome abnormalities etc. Chemicals analysed by Authors iniciated cancer development, but but further steps depend on many another factors: eg. genetic polymorphism of enzymes, metabolic pathways activity.

Experimental design

Analysis were done in small group of patients - anthors analysed results obtained from 37 individuals.
used methods for arsenic, cotinine and phthalates determination are proper and modern.

Validity of the findings

small group of patients involved in this study is main limitation of the study.

I agree with Authors conclusions. Phthalate metabolism may be is associated with cervical cancer development, but seeking for the cancer treatment it will be good to monitore estrogen levels.

Additional comments

in reviewed manuscript Authors analysed environmental exposure to arsenic, phthalates and tobacco smoke in relation locally advanced cervical cancer treatment.
In my opinion the idea that exposure in above factors in the past have impact on the progression of treatment is incorrect. These factors have impact on cancer development but in was no simple relations: eg. phthalates are endocrine disrupting chemicals and the way from phthalates exposure via changing hormonal metabolism to cancer development is very long; another mechanism are involved in arsenic induced carcinogenesis - it is observed increased oxidative stress active biomarkers, leading to chromosome abnormalities etc. Chemicals analysed by Authors iniciated cancer development, but but further steps depend on many another factors: eg. genetic polymorphism of enzymes, metabolic pathways activity, hormonal levels (estrogens).
l. 31 - Authors wrote: "... enrolled 43 patients...." and l. 191 ".....37 participants ..." does it mean that 6 patients were excluded from the study?

---

## Round 0.2 · Minor Revisions

Although the authors have addressed most of the issues raised by reviewers, there are still a few remaining that require additional consideration. Please address these and provide a point by point response as to how the issue was addressed and where the issue was addressed in the revised manuscript.

·

Basic reporting

The authors have responded to most of my previous suggestions. There are only some minor issues left:
The authors might have recruited 43 patients, but they only examined 37 because for the reminder no treatment outcomes were available yet. So maybe the figure (43) in the “methods” section of the abstract is misleading?
Although the authors acknowledge that they investigate cotinine as marker of nicotine exposure (both from active and from passive smoking) they still tend to write about “ETS” (environmental tobacco smoke, passive smoking, second hand smoke) which is not accurate. The highest cotinine levels are driven by active smoking, not by ETS. I suggest that in all instances when they write “ETS” (like in the title, in the abstract, and in many other instances) they consider to exchange this term with “tobacco exposure” or “nicotine intake” or “smoking/passive smoking” or whatever. I do believe this is would be more to the point.

Experimental design

this is a rather small sample. So the study is correctly termed a "pilot study" by the authors.

Validity of the findings

fine for me

Additional comments

I am looking forward to further analyses with a larger data set. I hope you will carry on!

---

## Round 0.3 · accepted · Accept

The issues raised by reviewers have been adequately addressed

·

Basic reporting

The paper is fine. From my point of view it can be published.
I am not sure about journal rules in that regard: I did not find any key-words in the submitted text.

Experimental design

This is but a pilot study. As such the design is valid.

Validity of the findings

Because of the small number of patients the findings should be considered as preliminary. This is clearly stated by the authors.

Additional comments

no comments